

**Effect of light on photosynthetic efficiency of sequestered**
**chloroplasts in intertidal benthic foraminifera (*Haynesina***
***germanica* and *Ammonia tepida*)**
**Thierry Jauffrais[1]\*, Bruno Jesus[2,3]\*, Edouard Metzger[1], Jean-Luc Mouget[4],**
**Frans Jorissen[1], Emmanuelle Geslin[1]**
[1]{UMR CNRS 6112 LPG-BIAF, Bio-Indicateurs Actuels et Fossiles, Université d'Angers,
2 Boulevard Lavoisier, 49045 Angers Cedex 1, France}
[2]{EA2160, Laboratoire Mer Molécules Santé, 2 rue de la Houssinière, Université de
Nantes, 44322 Nantes Cedex 3, France}
[3]{BioISI – Biosystems & Integrative Sciences Institute, Campo Grande University of
Lisboa, Faculty of Sciences, 1749-016 Lisboa, Portugal}
[4]{EA2160, Laboratoire Mer Molécules Santé, Université du Maine, Ave O. Messiaen,
72085 Le Mans cedex 9, France}
[\*]{The first two authors contributed equally to this work}.
Correspondence to: T. Jauffrais (thierry.jauffrais@univ-angers.fr)
**Abstract**
Some benthic foraminifera have the ability to incorporate functional chloroplasts from
diatoms (kleptoplasty). Our objective was to investigate chloroplast functionality of two
benthic foraminifera (*Haynesina germanica* and *Ammonia tepida*) exposed to different
irradiance levels (0, 25, 70 µmol photon m$^{-2}$ s$^{-1}$) using spectral reflectance, epifluorescence
observations, oxygen evolution and pulse amplitude modulated (PAM) fluorometry. Our
results clearly showed that *H. germanica* was capable of using its kleptoplasts for more than
one week while *A. tepida* showed very limited kleptoplastic ability with maximum
photosystem II quantum efficiency (*Fv/Fm* = 0.4), much lower than *H. germanica* and
decreasing to zero in only one day. Only *H. germanica* showed net oxygen production with a
compensation point at 24 µmol photon m$^{-2}$ s$^{-1}$ and a production up to 1000 pmol O$_2$ cell$^{-1}$ day$^{-1}$
$^{-1}$ at 300 µmol photon m$^{-2}$ s$^{-1}$. *Haynesina germanica Fv/Fm* slowly decreased from 0.65 to 0.55
in 7 days when kept in darkness; however, it quickly decreased to 0.2 under high light.



Kleptoplast functional time was thus estimated between 11 and 21 days in darkness and
between 7 and 8 days at high light. These results emphasize that studies about foraminifera
kleptoplasty must take into account light history. Additionally, this study showed that the
kleptoplasts are unlikely to be completely functional, thus requiring continuous chloroplast
resupply from foraminifera food source. The advantages of keeping functional chloroplasts
are discussed but more information is needed to better understand foraminifera feeding
strategies.
**1    Introduction**
Benthic foraminifera colonize a wide variety of sediments from brackish waters to deep-sea
environments and can be the dominant meiofauna in these ecosystems (Gooday 1986; Pascal
et al. 2009). They may play a relevant role in the carbon cycle in sediments from deep sea
(Moodley et al. 2002) to brackish environments (Thibault de Chanvalon et al. 2015). Their
secondary role in organic carbon cycling in aerobic sediments contrasts with their strong
contribution to anaerobic organic matter mineralisation (Geslin et al. 2011) and they can be
responsible for up to 80% of benthic denitrification (Pina-Ochoa et al. 2010; Risgaard-
Petersen et al. 2006). Some benthic foraminiferal species are known to sequester chloroplasts
from their food source and store them in their cytoplasm (Lopez 1979; Bernhard and Bowser,
1999) in a process known as kleptoplasty (Clark et al. 1990). A kleptoplast is thus a
chloroplast, functional or not, that was "stolen" and integrated by an organism. Kleptoplastic
foraminifera are found in intertidal sediments (e.g. *Haynesina*, *Elphidium* and *Xiphophaga*)
(Lopez 1979; Correia and Lee 2000, 2002a, b; Goldstein et al. 2010; Pillet et al. 2011), low
oxygenated aphotic environments (*Nonionella*, *Nonionellina*, *Stainforthia*) (Bernhard and
Bowser 1999; Grzymski et al. 2002) and shallow-water sediments (*Bulimina elegantissima*)
(Bernhard and Bowser, 1999).
The role of chloroplasts sequestered by benthic foraminifera is poorly known and
photosynthetic functions have only been studied in a few mudflat species (*Elphidium*
*williamsoni*, *Elphidium excavatum* and *Haynesina germanica*) (Lopez 1979; Cesbron pers.
comm.). Amongst the deep-sea benthic foraminifer living in the aphotic zone, only
*Nonionella stella* has been studied (Grzymski et al. 2002). The authors suggest that the
sequestered chloroplasts in this species may play a role in the assimilation of inorganic
nitrogen, even when light is absent. It has also been hypothesised that chloroplast retention
may play a major role in foraminiferal survival when facing starvation periods or in anoxic




environments (Cesbron pers. comm.). Under these conditions, kleptoplasts could potentially
be used as a carbohydrate source, and participate in inorganic nitrogen assimilation
(Falkowski and Raven 2007) or, when exposed to light, to produce oxygen needed in
foraminiferal aerobic respiration (Lopez 1979).
Foraminifera pigment and plastid ultrastructure studies have shown that the chloroplasts are
sequestered from their food source, i.e. mainly from diatoms (Lopez 1979; Knight and
Mantoura 1985; Grzymski et al, 2002; Goldstein 2004). This was confirmed by experimental
feeding studies (Correia and Lee 2002a; Austin et al. 2005) and by molecular analysis of
kleptoplastic foraminifera from different environments (Pillet et al. 2011, Tsuchiya et al.
2015). Foraminifera from intertidal mudflat environments (e.g. *H. germanica*, *A. tepida*) feed
mostly on pennate diatoms (Pillet et al. 2011) which are the dominant microalgae in intertidal
mudflat sediments (MacIntyre et al. 1996; Jesus et al. 2009). Furthermore, in this transitional
costal environments (e.g. estuaries, bays, lagoons) *A. tepida* and *H. germanica* are usually the
dominant meiofauna species in West Atlantic French coast mudflats (Debenay et al. 2000,
2006; Morvan et al. 2006; Bouchet et al. 2009; Pascal et al. 2009; Thibault de Chanvalon et
al. 2015). Their vertical distribution in the sediment is characterised by a clear maximum
density at the surface (Alve and Murray 2001; Bouchet et al. 2009; Thibault de Chanvalon et
al. 2015) with access to light, followed by a sharp decrease in the next two centimetres
(Thibault de Chanvalon et al., 2015).
Foraminiferal kleptoplast functional times can vary from days to months (Lopez 1979; Lee et
al. 1988; Correia and Lee 2002b; Grzymski et al. 2002). The source of this variation is poorly
known but longer kleptoplast functional times were found in dark treatments (Lopez 1979;
Correia and Lee 2002b), thus suggesting an effect of light exposure, similar to what is
observed in kleptoplastic sacoglossans (Trench et al. 1972; Clark et al. 1990; Evertsen et al.
2007; Vieira et al. 2009), possibly related to the absence of some components of the
kleptoplast photosynthetic protein complexes in the host (Eberhard et al. 2008).
Most recent studies on kleptoplastic foraminifera focused on feeding, genetics and
microscopic observation related to chloroplast acquisition (e.g., Austin et al. 2005, Pillet et al.
2011, Pillet and Pawlowski 2013). To our knowledge little is known about the effects of
abiotic factors on photosynthetic efficiency of sequestered chloroplasts in benthic
foraminifera, particularly on the effect of light intensity on kleptoplast functionality. Non-
invasive techniques are ideal to follow photosynthesis and some have already been used to





study foraminifera respiration and photosynthesis, e.g. oxygen evolution by microelectrodes
(Rink et al. 1998; Geslin et al. 2011) or $^{14}$C radiotracer (Lopez, 1979). Recently, pulse
amplitude modulated (PAM) fluorometry has been used extensively in the study of
kleptoplastic sacoglossans (Vieira et al. 2009; Costa et al. 2012; Jesus et al. 2010; Serodio et
al. 2010; Curtis et al. 2013; Ventura et al. 2013). This non-invasive technique has the
advantage of estimating relative electron transport rates (rETR) and photosystem II (PSII)
maximum quantum efficiencies (*Fv/Fm*) very quickly and without incubation periods. The
latter parameter has been shown to be a good parameter to estimate PSII functionality (e.g.
Vieira et al. 2009; Jesus et al. 2010; Serodio et al. 2010; Costa et al. 2012; Curtis et al. 2013;
Ventura et al. 2013).
The objective of the current work was to investigate the effect of irradiance levels on
photosynthetic efficiency and chloroplast functional times of two benthic foraminifera feeding
in the same brackish areas, *H. germanica*, which is known to sequester chloroplasts and *A.*
*tepida*, not known to sequester chloroplasts. These two species were exposed to different
irradiance levels during one week and chloroplast efficiency was measured using
epifluorescence, oxygen microsensors and PAM fluorometry.
**2  Materials and methods**
**2.1  Sampling**
*Haynesina germanica* and *A. tepida* were sampled in January 2015 in Bourgneuf Bay
(47.013°N, -2.019°W), a coastal bay with a large mudflat situated south of the Loire estuary
on the French west coast. In this area, all specimens of *A. tepida* belong to genotype T6 of
Hayward et al. (2004) (Schweizer pers. comm.). In the field, a large amount (±20 kg) of the
upper sediment layer (roughly first 5 mm) was sampled and sieved over 300 and 150 µm
meshes using *in situ* sea-water. The 150 µm fraction was collected in dark flasks and
maintained overnight in the dark at 18°C in the laboratory. No additional food was added. In
the following day, sediment with foraminifera was diluted with filtered (GFP, Whatman)
autoclaved sea-water (temperature: 18°C and salinity: 32) and *H. germanica* and *A. tepida* in
healthy conditions (i.e. with cytoplasm inside the test) were collected with a brush using a
stereomicroscope (Leica MZ 12.5). The selected specimens were rinsed several times using



Bourgneuf bay filtered-autoclaved seawater to minimize bacterial and microalgal
contamination.

## 2.2 Size and biovolume determination

Foraminifera test mean maximal elongation (μm) was measured using a micrometer mounted
on a Leica stereomicroscope (MZ 12.5). Mean foraminiferal volume was approximated with
the equation of a half sphere, which is the best resembling geometric shape for *H. germanica*
and *A. tepida* (Geslin et al. 2011). The cytoplasmic volume (or biovolume) was then
estimated by assuming that the internal test volume corresponds to 75% of the total
foraminiferal test volume (Hannah et al. 1994).

## 2.3 Spectral reflectance

Pigment spectral reflectance was measured non-invasively to determine the relative pigment
composition on 50 *H. germanica* and 50 *A. tepida* and a benthic diatom as explained in Jesus
et al. (2008). A USB2000 (Ocean Optics, Dunedin, FL, USA) spectroradiometer with a VIS-
NIR optical configuration controlled by OObase32 software (Ocean Optics B.V., Duiven, the
Netherlands) was used. The spectroradiometer sensor was positioned so that the surface was
always viewed from the nadir position. Foraminiferal reflectance spectra were calculated by
dividing the upwelling spectral radiance from the foraminifera (Lu) by the reflectance of a
clean polystyrene plate (Ld) for both of which the machine dark noise (Dn) was subtracted
(eq. 1).

$$\rho = \frac{(Lu - Dn)}{(Ld - Dn)} \qquad \text{(eq.1)}$$

## 2.4 Experimental design

*Haynesina germanica,* a species known to sequester chloroplasts, were placed in plastic Petri
dishes and starved during 7 days under three different light conditions: dark (D and Dark-
RLC, 3×10 foraminifera), low light (LL, 25 μmol photons m$^{-2}$ s$^{-1}$, 3×10 foraminifera) and
high light (HL, 70 μmol photons m$^{-2}$ s$^{-1}$, 3×10 foraminifera) on a 10:14 h (Light:Dark) cycle;
whereas for comparison, *A. tepida* (3×10 foraminifera), a foraminifer not known to sequester
chloroplasts were placed in plastic Petri dishes and only starved under dark conditions.



**2.5  Oxygen measurements**
Oxygen was measured at the beginning and end of the experiment using advanced Clark type
oxygen microelectrodes of 50 μm in diameter (Revsbech, 1989) (OXI50 - Unisense,
Denmark). Electrodes were calibrated with a solution of sodium ascorbate at 0.1 M (0%) and
with seawater saturated with oxygen by bubbling air (100%). Foraminiferal photosynthesis
and oxygen respiration rates were measured following Høgslund et al. (2008) and Geslin et al.
(2011). Measurements were carried out in a micro-tube made from glass Pasteur pipette tips
with an inner diameter of 1 mm. The micro-tube was fixed to a small vial, filled with filtered
autoclaved seawater from Bourgneuf Bay. The vial was placed in an aquarium with water
kept at room temperature (18°C). A small brush was used to position 7 to 10 foraminifera in
the glass micro-tube after removing air bubbles. Oxygen micro-profiles started at a distance of
200 μm above the foraminifers in the centre of the micro-tube and measurements were carried
out in 50 μm steps until 1000 μm away from the foraminifers (Geslin et al. 2011). For each
condition, three replicates were performed with different specimens. The oxygen flux (J) was
calculated using the first law of Fick:

$$J = -D \times \frac{dC}{dx} \qquad \text{(eq. 2)}$$

Where D is the oxygen diffusion coefficient (cm² s$^{-1}$) at experimental temperature (18°C) and
salinity (32) (Li and Gregory, 1974), and dC/dx is the oxygen concentration gradient (pmol
$O_2$ cm$^{-1}$). The $O_2$ concentration gradients were calculated using the oxygen profiles. Total $O_2$
consumption and production rates were calculated as the product of $O_2$ fluxes by the surface
area of the micro-tube and subsequently divided by the foraminifera number to finally obtain
the cell specific rate (pmol $O_2$ cell$^{-1}$ d$^{-1}$) (Geslin et al. 2011).
*Haynesina germanica* and *A. tepida* oxygen production and consumption were measured at
the beginning of the experiment using 3 replicates of 7 foraminifera each. Six different light
steps were used to measure $O_2$ production (0, 25, 50, 100, 200 and 300 μmol photons m$^{-2}$ s$^{-1}$)
for *H. germanica* and two light steps (0 and 300 μmol photons m$^{-2}$ s$^{-1}$) for *A. tepida*.
Photosynthetic activity (P) data of *H. germanica* were fitted with a Haldane model, as
modified by Papacek et al. (2010) and Marchetti et al. (2013) but without photoinhibition (eq.

29  3).

$$P(I) = \frac{Pm \times I}{I + Ek} - Rd \qquad \text{(eq. 3)}$$





Where Pm is the maximum photosynthetic capacity (pmol $O_2$ cell$^{-1}$ d$^{-1}$), I the photon flux density (μmol photons m$^{-2}$ s$^{-1}$), Ek the half-saturation constant (μmol photons m$^{-2}$ s$^{-1}$) and Rd the dark respiration, expressed as an oxygen consumption (pmol $O_2$ cell$^{-1}$ d$^{-1}$). The initial slope of the P–I (Photosynthesis –Irradiance) curve at limiting irradiance α (pmol $O_2$ cell$^{-1}$ day$^{-1}$ (μmol photons m$^{-2}$ s$^{-1}$)$^{-1}$)) and the compensation irradiance Ic were calculated according to equations 4 and 5.

$$Ic = \frac{Ek \times Rd}{Pm - Rd} \qquad \text{(eq. 4)}$$

$$\alpha = \frac{Rd}{Ic} \qquad \text{(eq. 5)}$$

Oxygen measurements were repeated at 300 μmol photons m$^{-2}$ s$^{-1}$ at the end of the experiment (7 days of incubation) for all different light treatments (D, LL, HL) to assess the production or consumption of oxygen at this light level.

## 2.6   Image analysis

*Haynesina germanica* kleptoplast fluorescence was measured using epifluorescence microscopy (×200, Olympus Ax70 with Olympus U-RFL-T) before and after the different light treatments. Two Tif images (1232 × 964 px) of each foraminifer (n = 30 per condition) were taken (one bright field photography and one epifluorescence photography) using LUCIA G$^{TM}$ software. The bright field photography was used to trace the contours of the foraminifer and an ImageJ macro was used to extract the mean pixel values of the corresponding epifluorescence photography. Higher mean pixel values corresponded to foraminifera emitting more fluorescence and thus, as a proxy, contain more chlorophyll. This was also measured on *A. tepida*, but results are not presented because no chlorophyll fluorescence was observed at the end of the experiment.

## 2.7   Fluorescence

All pulse amplitude modulated fluorescence measurements were carried out with a Water PAM fluorometer (Walz, Germany) using a blue measuring light. Chloroplast functionality was estimated using P-I rapid light curves (RLC, e.g., Perkins et al. (2006)) parameters (α, initial slope of the RLC at limiting irradiance; rETRmax, maximum relative electron transport rate; Ek, light saturation coefficient; and Eopt, optimum light) (Platt et al. 1980) and by



monitoring PSII maximum quantum efficiency (*Fv/Fm*). Rapid light curves were constructed
using eight incremental light steps (0, 4, 15, 20, 36, 48, 64, 90 and 128 µmol photons m$^{-2}$ s$^{-1}$),
each lasting 30 seconds. The PAM probe was set up on a stand holder at a 2 mm distance
from the foraminifera. *Fv/Fm* was measured daily at early afternoon, after a one-hour dark
adaptation period. All conditions (D, LL, HL and Dark-RLC) were done in triplicate. Rapid
light curves were carried out in all light treatments at the beginning and end of the
experiment, after one-hour dark adaptation for the 2 tested species. Additionally, RLC were
also carried out daily in one extra triplicate kept in the dark (Dark-RLC) throughout the
duration of the experiment (3×10 foraminifera).
**2.8   Statistical analysis**
Data are expressed as mean ± standard deviation (SD) when n = 3 or standard error (SE)
when n = 30. Statistical analyses consisted of a t-test to compare the foraminifera test mean
maximal elongation, a non parametric test (Kruskal Wallis) to compare the mean chlorophyll
fluorescence of the foraminifera exposed to the different experimental conditions and a
multifactor (experimental conditions (D, LL, HL), irradiance (0-300 µmol photons m$^{-2}$ s$^{-1}$))
analysis of variance (ANOVA) with a Fisher's LSD test to compare the respiration rates at the
end of the experiment. Differences were considered significant at $p<0.05$. Statistical analyses
were carried out using the Statgraphics Centurion XV.I (StatPoint Technologies, Inc.)
software.
**3    Results**
**3.1   Size and biovolume**
*Ammonia tepida* specimens were larger than *H. germanica* with a mean maximal elongation
of 390 µm (n = 34 and SD = 42 µm) and 366 µm (n = 122 and SD = 45 µm), respectively (p <
0.01, $F_{121,33}$ = 1.15). This resulted in cytoplasmic biovolumes equal to $1.20 \times 10^7$ µm$^3$ (SD =
$3.9 \times 10^6$ µm$^3$) and $1.01 \times 10^7$ µm$^3$ (SD = $3.65 \times 10^6$ µm$^3$).
**3.2   Chloroplast functionality**
*Haynesina germanica* and *A. tepida* showed very different spectral reflectance signatures
(Figure 1). *Haynesina germanica* showed a typical diatom spectral signature with high
reflectance in the infrared region (>740 nm) and deep absorption features around 435, 585,





630 and 675 nm; the absorption features around 435 and 675 nm correspond to the presence
of chlorophyll *a*; the 585 nm feature is the result of fucoxanthin and the 630 nm absorption
feature is the result of chlorophyll *c* (arrows, Figure 1). *Ammonia tepida* showed no obvious
pigment absorption features apart from 430 nm (Figure 1).
Epifluorescence images showed a clear effect of the different light treatments (Dark, Low
Light, Hight Light) on foraminiferal chlorophyll fluorescence (Figure 2). Visual observations
showed a clear decrease in chlorophyll fluorescence for the LL and HL treatments from the
beginning of the experiment (Figure 2A) to the end of a 7 day period of light exposure (Figure
2C and 2D, respectively). Samples kept in the dark did not show an obvious decrease but
showed a more patchy distribution compared to the beginning of the experiment (Figure 2B).
This was confirmed by a non-parametric test (Kruskal Wallis) showing that the differences in
chlorophyll *a* fluorescence were significant ($p < 0.01$, Df = 3, Figure 3). It is also noteworthy
to mention that there was a large individual variability within each treatment leading to large
standard errors in spite of the number of replicates (n = 30).
Oxygen measurements carried out at the beginning of the experiment (T0) differed
considerably between the two species. *Ammonia tepida* did not show any net oxygen
production although respiration rates measured at 300 μmol photons $m^{-2}$ $s^{-1}$ were lower (2485
± 245 pmol $O_2$ $cell^{-1}$ $d^{-1}$) than the ones measured in the dark (3531 ± 128 pmol $O_2$ $cell^{-1}$ $d^{-1}$)
($F_{2,2}$ = 3.7, $p = 0.02$). *Haynesina germanica* showed lower dark respiration rates (1654 ± 785
pmol $O_2$ $cell^{-1}$ $d^{-1}$) and oxygen production quickly increased with irradiance, showing no
evidence of photoinhibition (Figure 4). Compensation irradiance (Ic) was reached very
quickly, as low as 24 μmol photons $m^{-2}$ $s^{-1}$ (95% coefficient bound: 17-30 μmol photons $m^{-2}$ $s^{-1}$
$^{1}$, values calculated from the fitted model eq.4) and the half-saturation constant (Ek) was also
reached at very low light levels, i.e. at 17 μmol photons $m^{-2}$ $s^{-1}$. No photoinhibition was
observed under the experimental light conditions (0 to 300 μmol photons $m^{-2}$ $s^{-1}$), which
resulted in an estimation of ~2800 pmol $O_2$ $cell^{-1}$ $d^{-1}$ for maximum photosynthetic capacity.
The P-I curve initial slope at limiting irradiance (α) was estimated at 70 pmol $O_2$ $cell^{-1}$ $d^{-1}$
(μmol photons $m^{-2}$ $s^{-1}$)$^{-1}$ (95% coefficient bound: 58-88).
Oxygen measurements carried out at the end of the experiment (T7) showed significant
different dark and light respiration rates, with light respiration being lower than dark
respiration but not reaching net oxygen production rates (D, LL, HL) (Table 1). Moreover,
respiration rates were different between conditions ($p < 0.001$), with significantly lower




respiration rates of specimens incubated under High Light conditions than those under Dark
and Low Light conditions (p < 0.05, Fisher's LSD test.
PAM fluorescence rapid light curve (RLC) parameters (α, rETRmax, Ek and Eopt) showed
significant differences between foraminiferal species and over the duration of the experiment
(Figures 5 and 6). Highest rETRmax, α and Eopt were always observed in *H. germanica.*
After only one starvation day *A. tepida* RLC parameters dropped to zero or close to zero.
Contrastively, *H. germanica* RLC parameters showed a slow decrease throughout the
experiment (Figures 5 and 6) with rETRmax and α decreasing from 6 to 4 and 0.22 to 0.15,
respectively (Figures 6A and B). The parameters Ek and Eopt stayed constant over the 7 days
of the experiment, with values oscillating around 30 and 90, respectively (Figures 6C and D).
PSII maximum quantum yields (*Fv/Fm*) were clearly affected by light and time (Figure 7).
Both species showed high initial *Fv/Fm* values, i.e. > 0.6 and 0.4 for *H. germanica* and *A.*
*tepida,* respectively (Figure 7)*.* However*, w*hile *A. tepida Fv/Fm* values quickly decreased to
zero after only one starvation day, *H. germanica* exhibited a large variability between light
conditions (D, LL, HL) throughout the duration of the experiment (Figure 7); decreasing from
0.65 to 0.55 in darkness (D), from 0.65 to 0.35 under low light (LL) conditions and from 0.65
to 0.20 under high light (HL). Using these *Fv/Fm* decreases, *H. germanica* kleptoplast
functional times were estimated between 11-21 days in the dark (D), 9-12 days in low light
(LL) and 7-8 days in high light (HL); depending if an exponential or linear model was
applied. *Ammonia tepida* chloroplast functional times were estimated between 1-2 days
(exponential and linear model, respectively) and light exposure reduced the functional time to
less than one day (data not shown).

## 4  Discussion

### 4.1  Chloroplast functionality

Our results clearly show than only *H. germanica* was capable of carrying out net
photosynthesis. *Haynesina germanica* had typical diatom reflectance spectra (Figure 1),
showing the three major diatom pigment absorption features: chlorophyll *a*, chlorophyll *c*, and
fucoxanthin (Meleder et al. 2003; Jesus et al. 2008; Kazemipour et al. 2012; Meleder et al.
2013). Conversely, in *A. tepida* these absorption features were not detected, suggesting that





diatom pigments ingested by this species were quickly digested and degraded to a degree
where they were no longer detected by spectral reflectance measurements. These non-
destructive reflectance measurements are thus in accordance with other studies on benthic
foraminifera pigments by HPLC showing that *H. germanica* feed on benthic diatoms (Knight
and Mantoura, 1985). Similarly, Knight and Mantoura (1985) also detected higher
concentrations and less degraded diatom pigments in *H. germanica* than in *A. tepida.*
Furthermore, *H. germanica* has the ability to capture photons and produce oxygen from low
to relatively high irradiance, as shown by the low compensation point (Ic) of 25 μmol photons
$m^{-2}$ $s^{-1}$ and the high onset of light saturation (>300 μmol photons $m^{-2}$ $s^{-1}$) (Figure 4). Thus, *H.*
*germanica* seems to be well adapted to cope with the high light variability observed in
intertidal sediments that can range from very high irradiance levels during low tide to very
low levels within the sediment matrix or during high tide in turbid mudflat waters. *Ammonia*
*tepida* was found to carry out aerobic respiration, but respiration rates measured at 300 μmol
photons $m^{-2}$ $s^{-1}$ were lower than those measured in the dark. We thus suppose that in *A. tepida*
oxygen production by ingested diatom or chloroplasts might be possible, provided that this
species is constantly supplied with fresh diatoms. However, another possibility to explain this
reduction in oxygen consumption could be a decrease of its metabolism or activity under light
exposure. The light and dark oxygen production or consumption values measured for both
species are in accordance with previous studies (Geslin et al. 2011).
According to Lopez (1979), measured oxygen data can be used to estimate *H. germanica*
carbon fixation rates. Thus, using 1000 pmol $O_2$ $cell^{-1}$ $d^{-1}$ at 300 μmol photons $m^{-2}$ $s^{-1}$, ~200 to
4000 cells per 50 $cm^3$ in the top 0.5 cm (Morvan et al. 2006; Bouchet et al. 2007) and
assuming that photosynthesis produced one mol $O_2$ per mol of C fixed, *H. germanica* primary
production would be between $1.8 \times 10^{-5}$ and $4.0 \times 10^{-4}$ mol C $m^{-2}$ $d^{-1}$. This is a very low value
compared to microphytobenthos primary production in Atlantic mudflat ecosystems, which
usually range from 1.5 to 5.9 mol C $m^{-2}$ $d^{-1}$ (e.g. Brotas and Catarino 1995, reviewed in
MacIntyre et al. 1996). The estimated values represent thus less than 0.1% of
microphytobenthos fixated carbon and are in the same range of values than what has been
described by Lopez (1979) using [14]C radioactive tracers. These results should be interpreted
with caution because a wide variety of factors probably affect *H. germanica in situ* primary
production, e.g. diatom availability, kleptoplast densities, nutrient supply, light exposure, sea
water turbidity and migration capability are all factors that can potentially affect *H.*



*germanica* kleptoplast functionality. Nevertheless, although carbon fixation seems not to be
relevant at a global scale, the oxygen production could be important at a microscale and
relevant in local mineralization processes in/on mudflat sediments (e.g. iron, ammonium,
manganese).
At sampling time (T0) *H. germanica* rETR and *Fv/Fm* values were similar to
microphytobenthic species (i.e. *Fv/Fm* > 0.65) (Perkins et al. 2001), suggesting that the
kleptoplast PSII and electron transport chain were little affected after incorporation in the
foraminifers' cytoplasm. In contrast, *A. tepida Fv/Fm* and RLC parameters were already
much lower on the sampling day and quickly decreased to almost zero within 24 hours,
suggesting that plastids were not stable inside the *A. tepida* cytoplasm. Complete diatoms
inside *A. tepida* were already observed in feeding studies (Le Kieffre, pers. com), this low
*Fv/Fm* value might thus come from recently ingested diatoms by *A. tepida*. *Fv/Fm* has
previously been used to determine kleptoplast functional times and to follow decrease in
kleptoplast efficiency in other kleptoplastic organisms, e.g. the sea slug *Elysia virid*is (Vieira
et al. 2009). *Fv/Fm* measurements carried out on *H. germanica* at different light conditions
showed that light had a significant effect on the estimation of kleptoplast functional time, with
the longest functional time estimated at 21 days for dark condition. This time frame would
qualify *H. germanica* as a long term kleptoplast retention species (Clark et al. 1990);
however, our seven days estimation for the high light treatment would place *H. germanica* in
the medium-term retention group. This clearly shows that light exposure has an important
effect on this species kleptoplast functionality. Concerning *A. tepida,* the short dark diatom or
chloroplast functional time (<2 days) places this species directly in the short or medium-term
retention group.
Additionally, *H. germanica* kept in darkness showed a slow decrease of the RLC parameters,
α and rETRmax, throughout the seven experimental days; this decrease is likely related to
overall degradation of the light-harvesting complexes and of other components of the
photosynthetic apparatus, which gradually induced a reduction of light harvesting efficiency
and of carbon metabolism. This decrease was much amplified in low and high irradiance and
it should be pointed out that the actual light level of the HL treatment (i.e. 70 µmol photons
$m^{-2} s^{-1}$) is very low as compared to irradiances in their natural environment, which are easily
going above 1000 µmol photons $m^{-2} s^{-1}$, showing that the foraminifera kleptoplasts lack the
high photoregulation capacity exhibited by the benthic diatoms that they feed upon





(Cartaxana et al. 2013). This is consistent with the observation at the end of the experiment
that no net oxygen production was occurring under the different light conditions.
Nevertheless, a small difference was still found between dark and light respiration (Table 1),
suggesting that some oxygen production was still occurring but it was not sufficient to
compensate for the respiration oxygen consumption. We also noticed that the respiration was
higher in the foraminifera maintained in low light and dark conditions in comparison to the
high light foraminifera. In the line of the lower *Fv/Fm* values observed, this suggests that
kleptoplasts and possibly other metabolic pathways might have been damaged by the excess
of light. Clearly, in *H. germanica* light exposure had a strong effect on PSII maximum
quantum efficiency and on the retention of functional kleptoplasts (Figure 7), which can
explain the absence of net oxygen production after the 7 days of the experiments. Comparable
results for *H. germanica* were also obtained by counting the number of chloroplasts over time
with cells exposed or not to light (Lopez 1979). One of the most probable explanations for the
observed *Fv/Fm* decrease is the gradual inactivation of the protein D1 in PSII reaction
centres. This protein is an essential component in the electron transport chain and its turnover
rate is frequently the limiting factor in PSII repair rates (reviewed in Campbell and Tyystjärvi
2012). Normally, protein D1 is encoded in the chloroplast and is rapidly degraded and
resynthesized under light exposure with a turnover correlated to irradiance (Tyystjärvi and
Aro 1996). However, although D1 is encoded by the chloroplast genome, its synthesis and
concomitant PSII recovery require further proteins that are encoded by the algal nuclear
genome (Yamaguchi et al. 2005). Thus, when D1 turnover is impaired it will induce an *Fv/Fm*
decrease correlated to irradiance (Tyystjärvi and Aro 1996) consistent to what was observed
in the present study. In another deep sea benthic species (*Nonionella stella*) the D1 and other
plastid proteins (RuBisCO and FCP complex) were still present in the foraminifer one year
after sampling (Grzymski et al. 2002). This shows that some foraminifera can retain both
nuclear (FCP) and chloroplast (D1 and RuBisCO) encoded proteins. However, contrary to *H.*
*germanic*a, *N. stell*a lives in deeper environments never exposed to light and thus is unlikely
to carry out oxygenic photosynthesis (Grzymski et al. 2002). This fundamental difference
could explain why kleptoplast functional times are much longer in *N. stell*a, reaching up to
one year in specimens kept in darkness (Grzymski et al. 2002). On the other hand, it has been
shown that isolated chloroplasts are able to function for several months in Sacoglossan sea
slugs provided with air and light in aquaria (Green et al. 2001; Rumpho et al. 2001), which



demonstrates the existence of interactions between the kleptoplast and the host genomes, and
of mechanisms facilitating and supporting such long-lasting associations.
**4.2   Possible advantages of kleptoplasty for intertidal benthic foraminifera**
Much is still unknown about the relationship between kleptoplastic benthic foraminifera and
their sequestered chloroplasts. The relevance of the photosynthetic metabolism compared to
predation or organic matter assimilation is unknown; however, it would be of great interest to
understand the kleptoplast role in the foraminiferal total energy budget. Oxygenic
photosynthesis comprises multiple reactions leading to the transformation of inorganic carbon
to carbohydrates. However, to produce these carbohydrates all the light driven reactions have
to be carried out, as well as the Calvin cycle reactions. With fresh kleptoplasts this hypothesis
seems possible (e.g. Lopez 1979), especially if the plastid proteins are still present and
functional. However, we showed that the maximum quantum efficiency of the PSII decreased
quickly under light exposure, suggesting that substantial direct carbohydrate production is
unlikely without constant chloroplast replacement. Conversely, the production of intermediate
photosynthetate products such as adenosine triphosphate (ATP) and nicotinamide adenine
dinucleotide phosphate (NADPH) could be possible and would be of metabolic value for the
foraminifera. It is also possible that *in situ* the foraminifera have better photoregulation
capacities. Not only they will have easy access to fresh diatom chloroplasts, as *H. germanica*
is mainly living in the first few mm of the superficial sediment (Alve and Murray 2001,
Thibault de Chanvalon et al. 2015), but they will also have the possibility of migrating within
the sediment (Gross 2000) using this behavioural feature to enhance their photoregulation
capacity, similarly to what is observed in benthic diatoms from microphytobenthic biofilms
(e.g. Jesus et al. 2006; Mouget et al. 2008; Perkins et al. 2010). However, below the photic
limit (max 2 to 3 mm in estuarine sediments (reviewed in MacIntyre et al. 1996, Cartaxana et
al. 2011)) it is unlikely that oxygenic photosynthesis will occur, and live *H. germanica* are
also found below this limit (Thibault de Chanvalon et al. 2015).
Using kleptoplasts, *H. germanica,* like other kleptoplastic organisms (e.g. *Elysia viridis*
(Teugels et al. 2008)), is also theoretically capable of assimilating inorganic nitrogen via the
glutamine synthetase and glutamate 2-oxo-glutarate aminotransferase (GS-GOGAT)
pathways to produce glutamate and glutamine after the successive reduction of nitrate to
nitrite and nitrite to ammonia or directly through ammonium uptake (Zehr and Falkowski
1988). However, the first reduction occurs in the diatom cytoplasm via the enzyme nitrate



reductase (NR) and not inside the chloroplast. It is not known if *H. germanica* has this
enzyme but it is present in *N. stella* (Grzymski et al. 2002). Interestingly, nitrogen (i.e. nitrite
and ammonium) assimilation by sacoglossans (e.g. *Elysia viridis*) was observed under light
and dark conditions with significantly higher nitrogen assimilation observed under light
condition (Teugels et al. 2008). The uptake of ammonium and nitrite are light dependent as
their relevant enzymes require kleptoplast electron donors (NiR and GOGAT); i.e. reduced
ferredoxin formed in the photosynthetic electron transport chain are used as electron donors in
the reaction involving the nitrite reductase [NiR]. Furthermore, the GS metabolic reaction is
ATP-dependent, and gene expression of some key enzymes (NiR, GS and GOGAT) is light
regulated (Grossman and Takahashi 2001). This suggests that kleptoplasts might also have an
added value in providing extra nitrogen source to metabolic pathways in foraminifera under
light exposure and also possibly over short periods under dark conditions. It is also
noteworthy that ammonium incorporation might take place through the glutamine
dehydrogenase (GDH) pathway in the mitochondria that converts glutamate to α-
ketoglutarate, which can subsequently be assimilated in the kleptoplast via the GOGAT
pathway (Teugels et al. 2008).
Diatoms are also known to assimilate organic nitrogen (Antia et al. 1991), to use their
ornithine-urea cycle for anaplerotic carbon fixation into nitrogenous compounds (Allen et al.
2012) and some of the benthic species present on mudflats are also able to assimilate organic
carbon (Admiraal and Peletier 1979). Apparently some benthic diatoms can alternate between
an auto- or heterotrophic metabolism in function of the environment. Analysing the
kleptoplast DNA would provide interesting data to determine if foraminifera are capable of
selecting facultative heterotrophic diatoms to improve their ability to assimilate dissolved
organic compounds. Finally, another possible added value of incorporating kleptoplasts is the
possibility of using them as an energy stock to be digested during food-impoverished periods
particularly when foraminifera are transported below the photic zone of the sediment by
macrofaunal bioturbation.

## 5  Conclusion

Comparing *H. germanica* with *A. tepida* showed that the former species potentially has the
capacity of retaining functional kleptoplasts up to 21 days, much longer than *A. tepida* that
showed almost no PSII activity after 24 hours. Nevertheless, the capacity of *H. germanica* to
keep functional kleptoplasts was significantly decreased by exposing it even to low irradiance



levels, which resulted in low *Fv/Fm* values and decreased oxygen production. This shows
clearly that in our experimental conditions, *H. germanica* had reduced photoregulation
capacities. These results emphasize that studies on kleptoplast photophysiology of benthic
foraminifera must be interpreted with care, as results are strongly influenced by the
foraminiferal light history before incubation. Additionally, this study shows that the cellular
machinery necessary for chloroplast maintenance is unlikely to be completely functional,
suggesting that *H. germanica* has to continuously renew its chloroplasts to keep them
functional. We hypothesize that kleptoplasts might have an added value by providing extra
carbon and fueling nitrogen metabolic pathways to foraminifera, mainly under light exposure,
but also as energy stock to be digested during food impoverished periods, in dark or light
conditions.
**Acknowledgements**
This study is part of the EC2CO project "FORChlo" supported by the CNRS. This study is
strongly supported by the Region Pays de la Loire (Post-doc position of the first author and
"COSELMAR" and "Fresco" projects).

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



1    Table 1. Light and dark respiration rates (pmol $O_2$ cell$^{-1}$ d$^{-1}$) ± SD of *Haynesina germanica* in

2    the three experimental conditions (Dark, Low Light and High Light) at the end of the

3    experiment (Df, degree of freedom, PFD Photon Flux Density).

| Condition | PFD | Respiration Rate (pmol $O_2$ cell$^{-1}$ d$^{-1}$) | | |
|---|---|---|---|---|
| D | 300 | 2452 ± 537 | | |
| | 0 | 3542 ± 765 | | |
| LL | 300 | 3468 ± 305 | | |
| | 0 | 4015 ± 110 | | |
| HL | 300 | 1179 ± 261 | | |
| | 0 | 1905 ± 235 | | |
| Anova | | Df | F-test | p |
| Condition | p (α=0.05) | 2 | 13.1 | <0.001 |
| PFD | p (α=0.05) | 1 | 5.4 | 0.026 |
| Interaction | p (α=0.05) | 2 | 0.3 | 0.78 |



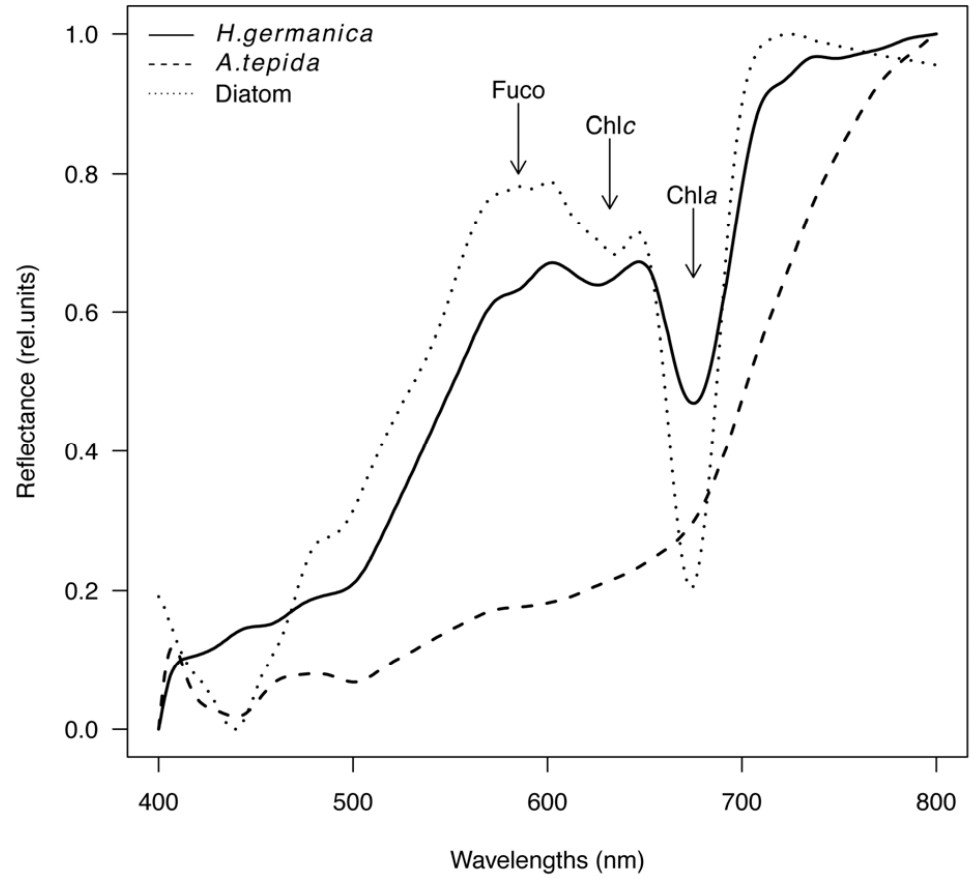

2     Figure 1. Spectral reflectance signatures of *Haynesina germanica, Ammonia tepida* and of a

3     benthic diatom in relative units (X-axis legend: Wavelength (nm)).



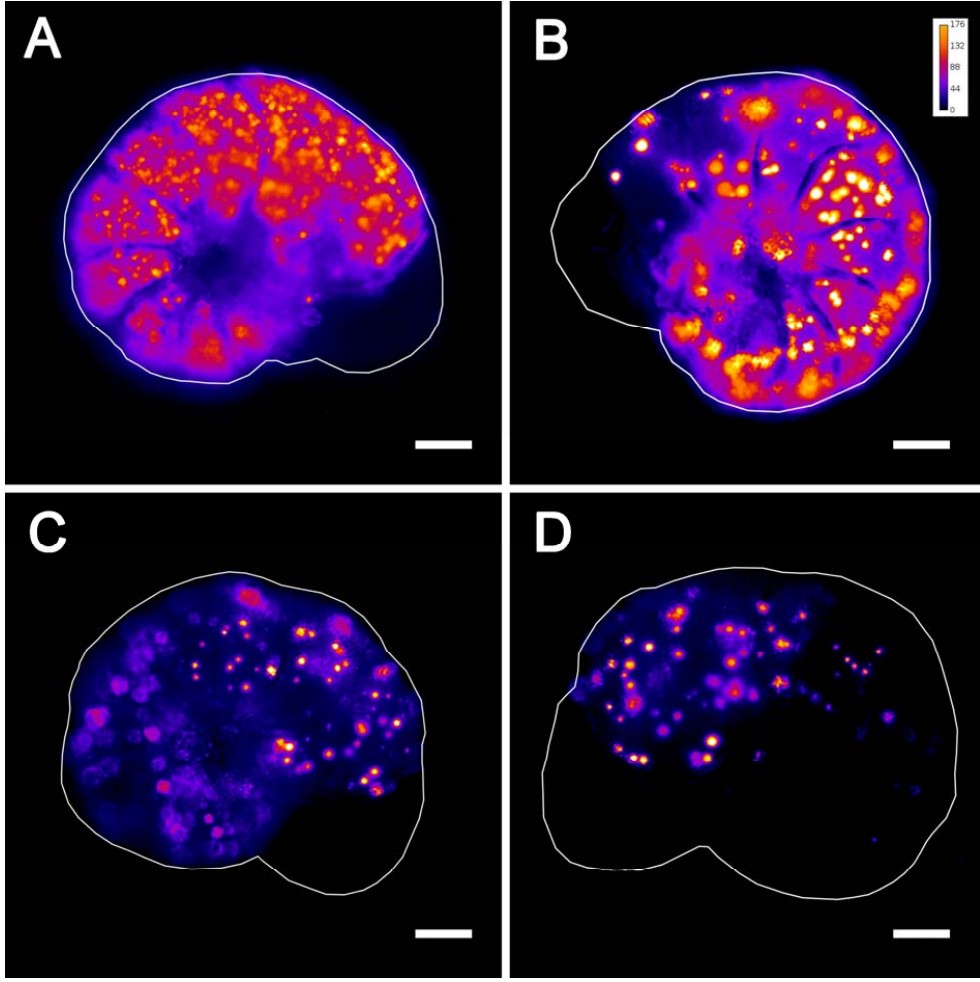

Figure 2. Illustration of *Haynesina germanica* chloroplast content at the beginning (A) and at the end of the experiment for the three experimental conditions, Dark (B), Low Light (C) and High Light (D). Higher colour scale values correspond to foraminifera emitting more fluorescence and likely containing more chlorophyll *a*; fluorescence in pixel values between 0 and 255, (scale bar = 50 μm).



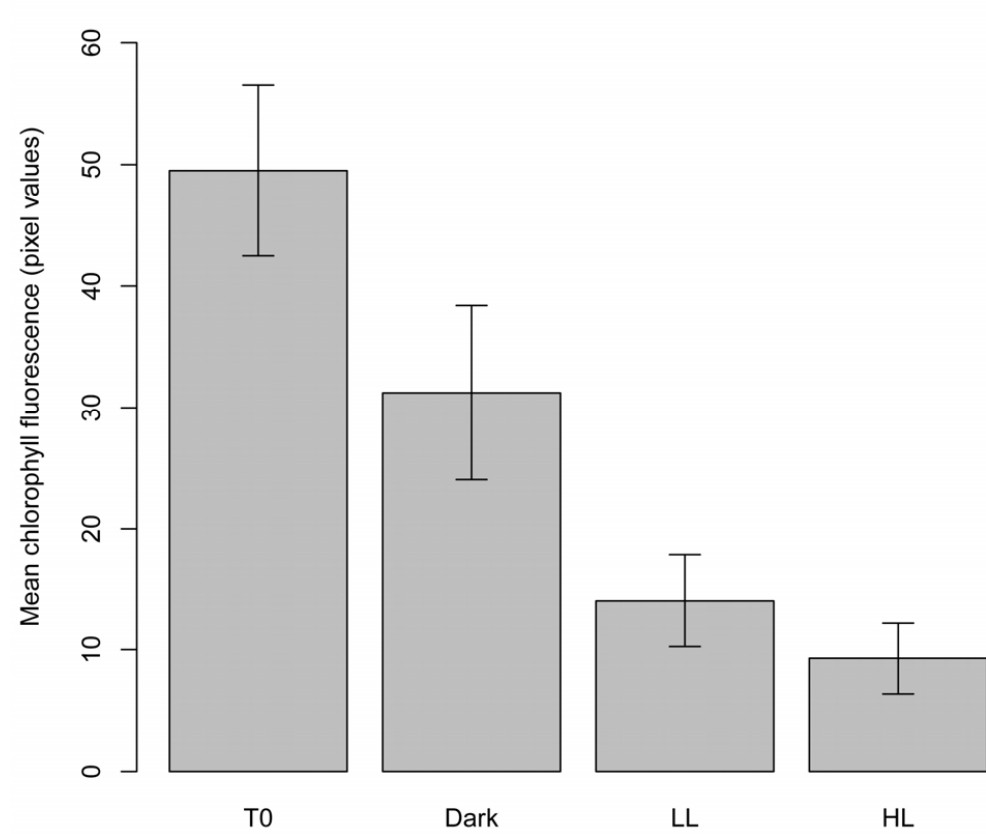

Figure 3. Mean chlorophyll *a* fluorescence (± SE, n = 30) at the end for the three experimental
conditions (Dark, Low Light and High Light) and the beginning (T0) of the experiment using
*Haynesina germanica*. Higher mean values likely corresponded to foraminifera containing
more chlorophyll.





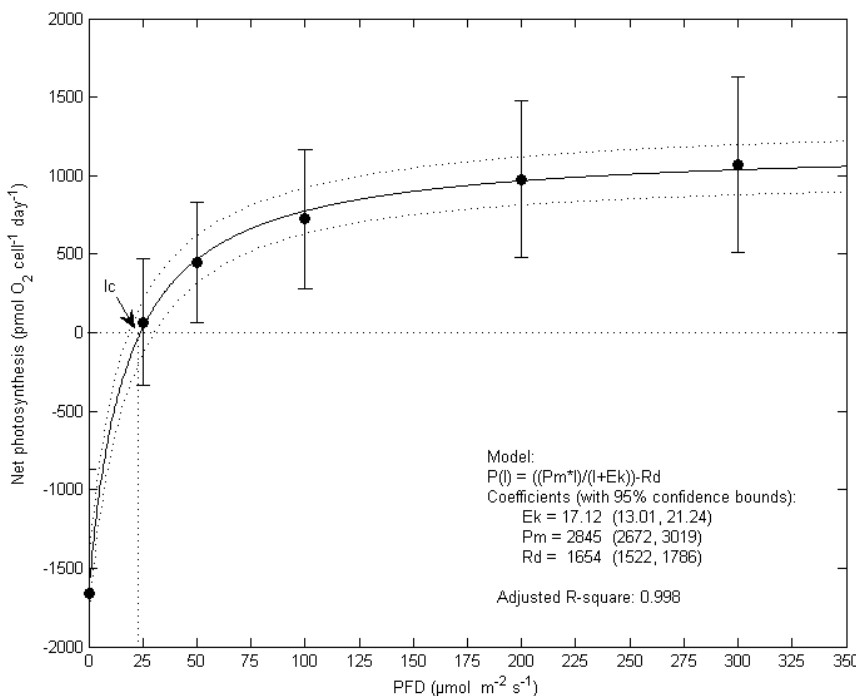

Figure 4. Net photosynthesis of *Haynesina germanica* (pmol $O_2$ cell$^{-1}$ d$^{-1}$) as a function of the photon flux density (PFD, μmol photons m$^{-2}$ s$^{-1}$). The half-saturation constant, Ek, was found at 17 (13-21), the dark respiration, Rd, at 1654 (1522-1786) pmol $O_2$ cell$^{-1}$ d$^{-1}$ and the maximum photosynthetic capacity, Pm, at 2845 (2672-3019) pmol $O_2$ cell$^{-1}$ d$^{-1}$. The Ic, calculated compensation irradiance (24 (17-30) μmol photons m$^{-2}$ s$^{-1}$). The adjusted R² of the model was equal to 0.998, n = 3.





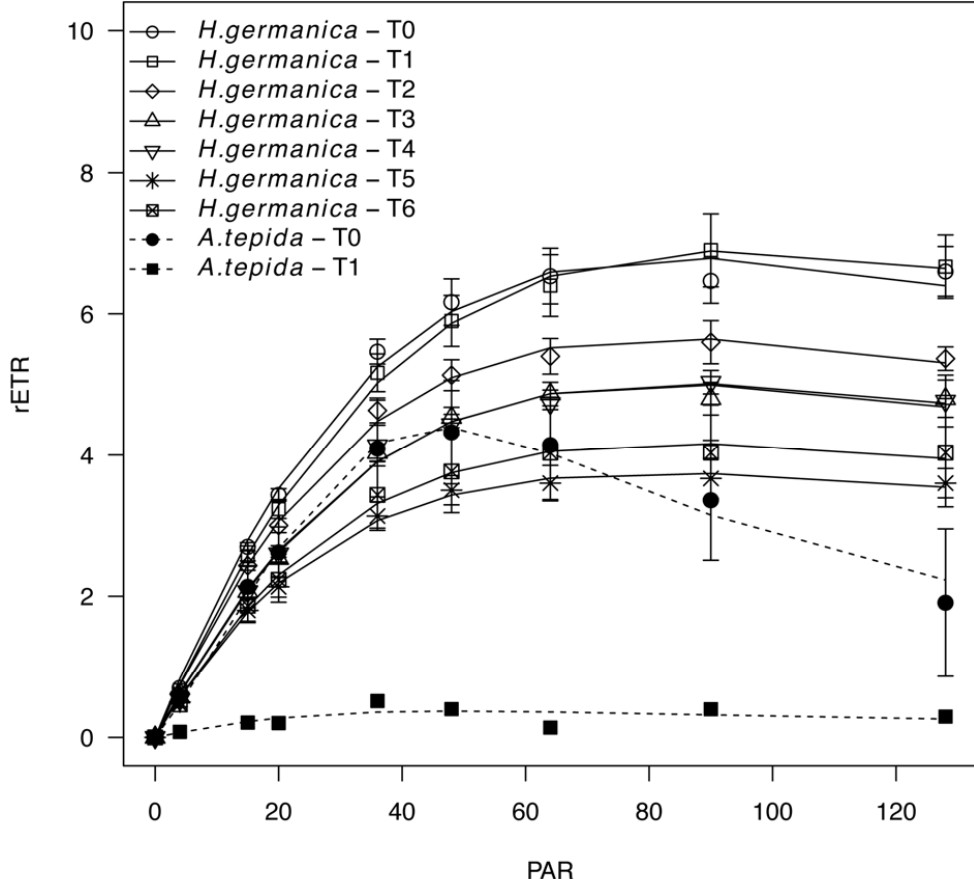

Figure 5. Rapid light curves (RLC, n = 3) expressed as the relative electron transport rate
(rETR) as a function of the photosynthetic active radiation (PAR in µmol photons $m^{-2}$ $s^{-1}$) of
*Haynesina germanica* (black lines) and *Ammonia tepida* (black dashed lines) during the seven
days of the experiment.





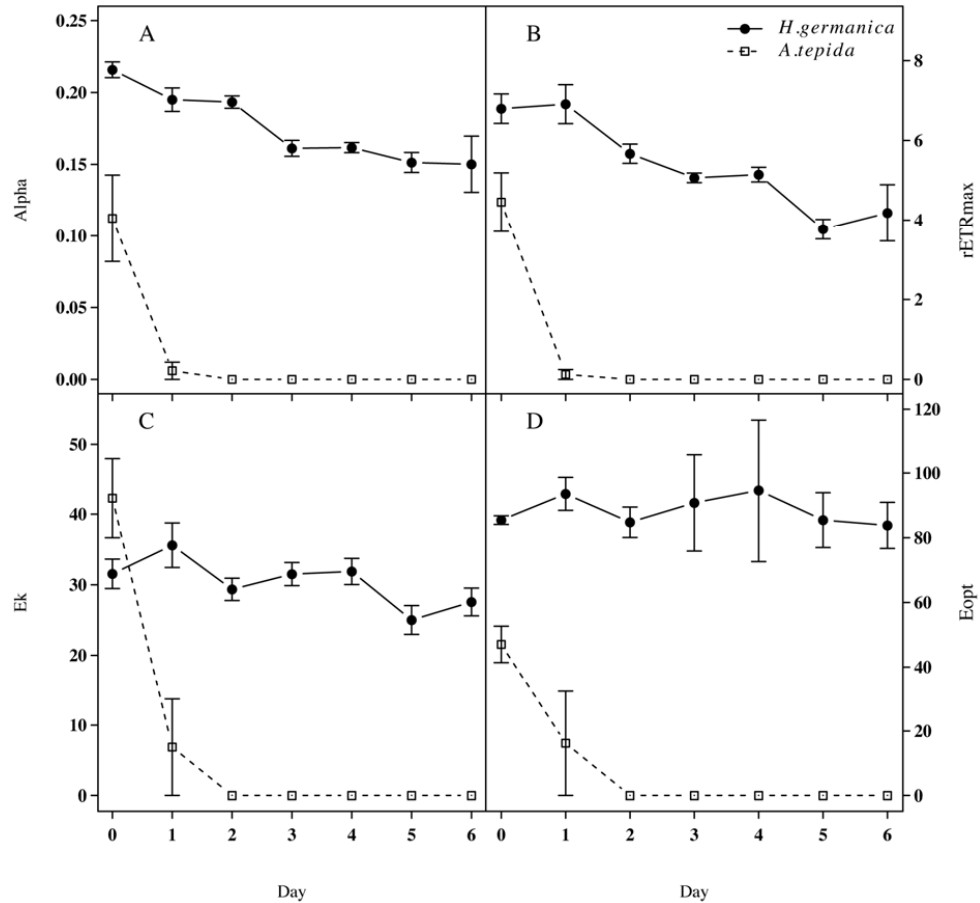

Figure 6. Rapid light curve (RLC, n = 3) parameters for *Haynesina germanica* (Dark-RLC)
and *Ammonia tepida* maintained in the dark during the experiment, Alpha is the initial slope
of the RLC at limiting irradiance, rETRmax is the maximum relative electron transport rate,
Ek is the light saturation coefficient and Eopt is the optimum light, all of them were estimated
by adjusting the experimental data to fit the model of Platt et al. (1980).





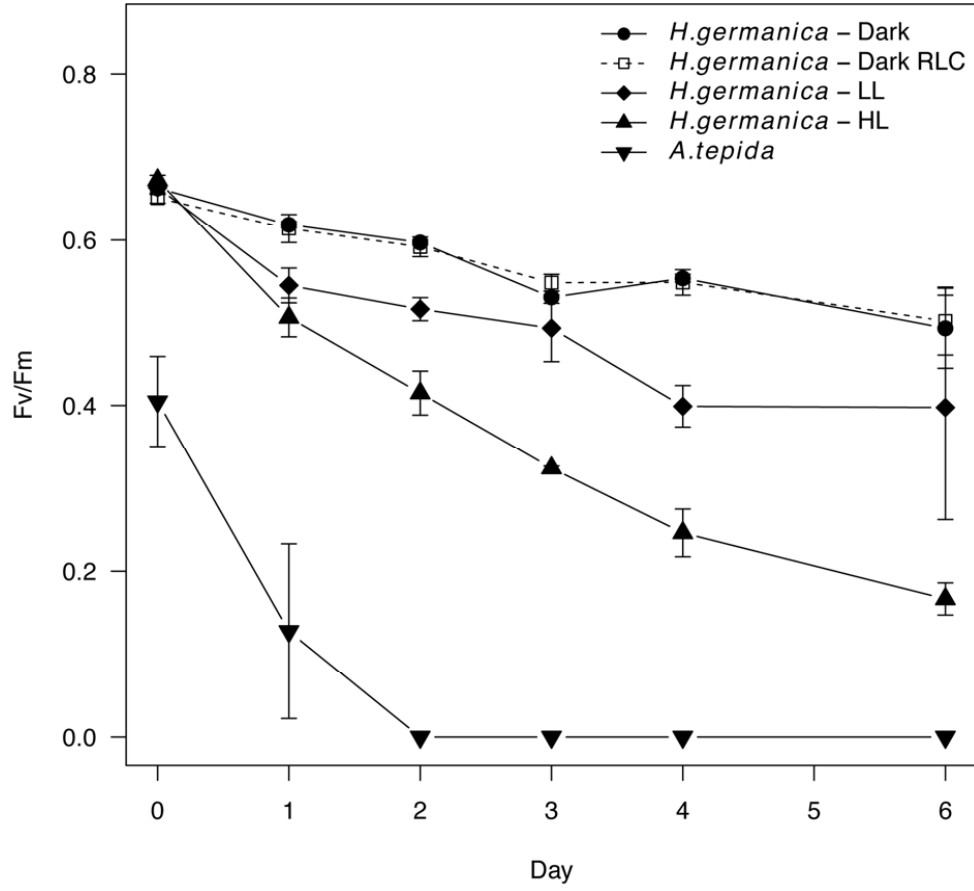

2 Figure 7. Maximum quantum efficiency of the photosystem II (*Fv/Fm*, n = 3) during the

3 experiment for the different applied conditions (Dark, Low Light and High Light) and species

4 (*Haynesina germanica* and *Ammonia tepida*).