# Peer review of "Effect of light on photosynthetic efficiency of sequestered"

_Biogeosciences, 2015_

## Referee Comment (RC1) · Anonymous Referee #1 · 12 Feb 2016

The manuscript reported about the effect of different light intensities on chlorophyll concentrations, photosynthetic capabilities, and oxygen production/consumption rates during 7 days incubation experiments. They found that chlorophyll concentrations and photosynthetic capabilities differ with light intensity, even between low level of light intensities. The authors also reported that A. tepida did not show such long retaining of chloroplast, suggesting H. germanica should have some way to keep chloroplast, not just digesting them. Some of the findings are new (and the method they used is probably new for foraminiferal kleptoplasty study), but the present manuscript should be re-organized before its publication.
[Figure]

Introduction In the introduction, the authors need to specify (or concentrate) more on precise objective of the authors study: i.e. what is known about the light intensity effects on kleptoplasty (only dark and light comparison before?), and why the authors need to clarify light intensity effects, not a function of chloroplast etc.

Discussion 4.2. Most of this section, in particular 2nd and 3rd paragraphs, the discussions are stretchs from the current manuscript. I am sure that the ecological role of kleptoplasty is very important topic and the authors' future goal would be this scope, however, the current manuscript reported about the effect of light intensity on the chlorophyll intensity (chloroplast abundances) and its photosynthetic efficiency. If the authors want to keep these discussions, they must discuss by incorporating their findings in this manuscript. I rather suggest to discuss about the meaning of the authors findings that the chlorophyll retention times and photosynthetic capabilities differ greatly between LL and HL, "although HL is still far below the natural photon radiation levels".

Other minor comments or corrections

Page 2 Line 12 If the authors mention "secondary role" here, then the authors need to mention about that bacteria play primary roles on carbon cycling in aerobic sediments.

Line 16 The sentence starting with "Some benthic foraminifera..." seems appeared abruptly. Is kleptoplasty related to carbon cycling or anoxic adaptation of the foraminifera? If so, please add relevant connections from the former sentences.

Page 3 Line 13 Costal > coastal

Page 4 Line 21 What is the "-" before 2.019W? Does this mean 2.019E?

Line 23 ~20 kg

Line 27 Please note the filter size

Page 5 Line 11 Please explain shortly about the methods described in Jesus et al.

(2008).

Line 12 50 specimens of H. germanica and . . .

Line 25 This is the first place appearing RLC, so please explain.

3 X 10 specimens (or individuals)

Page 6 Line 11 How long did the authors wait till the oxygen microprofiling after putting foraminifera into the tube?

Line 19 Which position of oxygen gradients were used to calculate diffusion flux? Near foraminifera? Maximum slope? Or did authors proximate in some way? Please specify and describe.

Page 7 Line 20 I guess Ammonia tepida exhibited chlorophyll at the start of the experiment because they still have some diatoms in food vacuoles. It may be interesting to compare the concentration of chloroplast at the begging (perhaps reflecting selective ingestion?) or reduction of chlorophyll in A. tepida as an index of degradation of chloroplast and that of H. germanica, which retain chloroplast.

Line 25 Please note wave length

Page 8 Line 5 Triplicate measurement for each specimen? or just using 3 specimens as triplicate? Please specify.

Line 9 Again, individuals or specimens are better than using "foraminifera"

Line 12 To compare the foraminifera test mean maximal elongation "between what"

Line 23 "390 +- 42 um (SD, n = 34)" is better

Line 29 Absorption at 435 and 585 nm are not "deep absorption feature".

Page 9 Line 1 In the figure 1, there is no indication of Chla at 435 nm wavelength.

Line 3 Please note which sample (starved and kept dark under 7 days?) was used for

this spectral signatures in Figure 1.

Line 12 There is no statistical indication in Figure 3. Also, Kruskal Wallis can detect differences between several samples, but cannot say anything about the difference between specific two samples. Therefore, if the authors describe "Samples kept in the dark did not show an obvious decrease", then the authors need to perform another statistical analysis on this.

Regarding figure 3, did the authors perform any kind of "calibration" between pixel values and chlorophyll concentration? If not, the vertical axis (pixel values) does not have any numerical meaning. I therefore suggest to present as relative chlorophyll fluorescence as T0=100%.

Line 21 No evidence of photoinhibition "of this measured range" or something

Line 30 "light respiration being lower than dark respiration" Based on the Table 1, LL respiration was higher than dark respiration. Does "light respiration" mean the average of LL and HL? Please specify.

Page 10 Line 2 LSD test ")".

Line 26 Clearly show "that"?

Page 11 Line 8 "24" umol photons?

Line 11 Do the authors have any idea on the in situ light intensity?

Page 12 Line 7 It seems the authors want to say "modestly" or something instead "little"

---

## Referee Comment (RC2) · Anonymous Referee #2 · 17 Mar 2016

Author analysed the functionally of chloroplast retained by some species of benthic foraminifera. Study conducted is very interesting, and the techniques used are new and applicable to other organisms, which makes the manuscript relevant to a broad readership. However, methods section needs to be carefully revised as it does not follow a logical sequence, and experimental design needs to be explained in more detail. Moreover, manuscript needs to be proofread and revised by a native English speaker. Many problems with punctuation throughout the text.

Introduction

Ln. 19-24: Kleptoplasty is also very common in carbonate reef environments when conditions are favourable (i.e, oligotrophy; e.g., Ziegler and Uthicke 2011).

Ln. 25-28: Studies by Correia and Lee need to be acknowledged and cited here as they represent a good contribution to this research field.

Methods

Ln. 11-16: Please provide a rationale for only exposing the specimens to different light levels for one week only.

Experimental design: Please, clarify the total number of individual used per replicate and number of replicates per treatment.

Ln. 27-28: Clarify why A. tepida specimens were not starved under light conditions, and if A. tepida was exposed to different light conditions at all.

Also, please clarify the experimental design. Was A. tepida exposed to different light treatments? There is no information in the methods (where it should be). It is surprising that the authors only used one paragraph to explain their experimental design, which is the most important part of the study. There is no way for the reader to know number of replicates, total number of specimens, why conditions were chosen, how light levels were reached, temperature, static or flow-through system? Detail explanation of the experimental design is necessary.

Methods section does not follow a logic sequence when explaining each parameter analysed. This section needs to be carefully revised.

Were all specimens used in the experiment tested for all parameters analysed? Please,

clarify.

Ln. 10-11: Was one individual used at a time or all at once? Please, clarify.

Ln. 20-22: What about inter specific differences? Did the authors use a pool of 7-10 individuals for O2 consumption measurements? Or the measurements were done individually?

Ln. 24: Authors stated that seven specimens were used, but previously (ln. 10) mentioned "7 to 10 foraminifera". Please, be consistent.

Ln. 26: Please clarify why only two steps were used for A. tepida.

Fluoresce measurements: What light was used to measure Fo? Please, clarify

Ln. 15-16: It seems that the authors have a blocked design, but it hard to tell based on the current description of the experimental design. For example, if both species were put in the same experimental petri dish or not. That requires a more detailed description of the methods. Therefore, it is impossible to judge if authors conducted the appropriate statistical analyses.

Throughout the methods section author put in brackets "3x10 foraminifera". Please, clarify if this means replicates or trials per parameter analysed.

Ln. 24-25 Please add ", respectively", after "This resulted in cytoplasmic biovolumes equal to $1.20 \times 107~\mu m3$ (SD = 25 $3.9 \times 106~\mu m3$) and $1.01 \times 107~\mu m3$ (SD = $3.65 \times 106~\mu m3$)"

Ln. 5-6: Figure 2 only shows data on H. germanica fluorescence. Please, amend the

sentence accordingly.

Ln. 15-19: The manuscript would improve if all these numbers were put in a table or graph.

Ln. 20-22: Please, clarify why data is not shown. Maybe authors could add these results to supplementary material, if possible.

The manuscript would benefit from a figure plotting the relative difference of Fv/Fm between light treatments, specially low and high light levels.

Ln. 7-12: Figure 4 does not show this result.

Ln. 21-23: This is expected, given that exposure to high light levels generates a lot of reactive oxygen species inside the chloroplasts. This should be mentioned and discussed.

Ln. 21-23: A. tepida has no capacity to retain chloroplast according to the results, as fluoresce only persists for a couple of days, and even though some fluorescence is detected, the functionality was not analysed. Therefore, chloroplasts might be present for a couple of days, but not functional. The O2 consumption is not a proxy of functionally of kleptoplasts, and just because respiration rates were lower at 300 uE does not mean that chloroplasts were functioning. Be careful not to mix up correlation with causation.

Ln. 28-32: This is very interesting. I wonder what caused this significant reduction in tolerance in these chloroplasts. Maybe the lack of a cellular protection? Would be great to see a sentence or two with thoughts from the authors of why such dramatic decrease. It would be possible that in situ the chloroplast are not functional at all.

Ln. 7-9: Chloroplasts are naturally hyperoxic and, as mentioned previously, produce reactive oxygen species, which make this organelles susceptible to oxidative stress. Reactive oxygen species in the chloroplast can cause damage to PS II, primarily through oxidative degradation of essential proteins. This is important to be added to the discussion. It would explain why Fv/Fm of H. germanica decreases with increases in light level.

Ln. 29-30: Please add "within the light range tested in this study": "Comparing H. germanica with A. tepida showed that the former species potentially has the 30 capacity of retaining functional kleptoplasts up to 21 days, within the light range tested in this study"

Figure 1: Please, mention the species of diatom and reference

Figure 4: Which treatment is plotted in the graph? As stated in the text, P-I curves were measured for all treatments (page 7, ln. 9-11). Please, clarify. It would be interesting to see the P-I curves of specimens exposed to all treatments.

Figure 5: Please add the letters (A, B, C and D) to the legend.

---

## Author Comment (AC1) · 22 Apr 2016

Dear Dr. Middelburg,

Please find below our response to the different comments done by the reviewers, we improved the material and methods section and reduced the discussion as requested.

We also enclosed our revised manuscript entitled "Effect of light on photosynthetic efficiency of intertidal benthic foraminifera" by Thierry Jauffrais, Bruno Jesus, Edouard Metzger, Jean-Luc Mouget, Frans Jorissen, Emmanuelle Geslin.

[Figure]

We hope that these revisions will fulfill all the requests and will make the manuscript acceptable for publication.

Thank you for your work, .

Best regards,

Thierry Jauffrais
* * *
Dr. Thierry Jauffrais UMR CNRS 6112 LPG-BIAF, Université d'Angers, UFR Sciences, 2 Bd Lavoisier, 49045 ANGERS CEDEX 01, France Thierry.jauffrais@univ-angers.fr TEL: +332 41 73 50 09

Response to reviewer

Anonymous Referee #1

 The manuscript reported about the effect of different light intensities on chlorophyll concentrations, photosynthetic capabilities, and oxygen production/consumption rates during 7 days incubation experiments. They found that chlorophyll concentrations and photosynthetic capabilities differ with light intensity, even between low level of light intensities. The authors also reported that A. tepida did not show such long retaining of chloroplast, suggesting H. germanica should have some way to keep chloroplast, not just digesting them. Some of the findings are new (and the method they used is probably new for foraminiferal kleptoplasty study), but the present manuscript should be re-organized before its publication.

Introduction Comment: In the introduction, the authors need to specify (or concentrate) more on precise objective of the authors study: i.e. what is known about the light intensity effects on kleptoplasty (only dark and light comparison before?), and why the authors need to clarify light intensity effects, not a function of chloroplast etc.

Reply: We politely disagree, we clearly state the objective in P4L12 and the introduction already contains the existing information about our topic, e.g. P3L20-27, P3L30

P3 line 21-27: "Foraminiferal kleptoplast retention times can vary from days to months (Lopez 1979; Lee et al. 1988; Correia and Lee 2002b; Grzymski et al. 2002). The source of this variation is poorly known but longer kleptoplast retention times were found in dark treatments (Lopez 1979; Correia and Lee 2002b), thus suggesting an effect of light exposure, similar to what is observed in kleptoplastic sacoglossans (Trench et al. 1972; Clark et al. 1990; Evertsen et al. 2007; Vieira et al. 2009), possibly related to the absence of some components of the kleptoplast photosynthetic protein complexes in the host (Eberhard et al. 2008)." P3 Line 30: "To our knowledge little is known about the effects of abiotic factors on photosynthetic efficiency of sequestered chloroplasts in benthic foraminifera, particularly on the effect of light intensity on kleptoplast functionality." P4 Line 12:" The objective of the current work was to investigate the effect of irradiance levels on photosynthetic efficiency and chloroplast functional times of two benthic foraminifera feeding in the same brackish areas, H. germanica, which is known to sequester chloroplasts and A. tepida, not known to sequester chloroplasts."

Discussion Comment: 4.2. Most of this section, in particular 2nd and 3rd paragraphs, the discussions are stretchs from the current manuscript. I am sure that the ecological role of kleptoplasty is very important topic and the authors' future goal would be this scope, however, the current manuscript reported about the effect of light intensity on the chlorophyll intensity (chloroplast abundances) and its photosynthetic efficiency. If the authors want to keep these discussions, they must discuss by incorporating their findings in this manuscript. I rather suggest to discuss about the meaning of the authors findings that the chlorophyll retention times and photosynthetic capabilities differ greatly between LL and HL, "although HL is still far below the natural photon radiation levels".

Reply: We agree, to make the manuscript more focused and within the scope of our experimental work we deleted from the reviewed manuscript the 2nd and 3rd paragraphs of the section 4.2: "Using kleptoplasts, H. germanica, like other kleptoplastic organisms (e.g. Elysia viridis (Teugels et al. 2008)), is also theoretically capable of assimilating inorganic nitrogen via the glutamine synthetase and glutamate 2-oxo-glutarate aminotransferase (GS-GOGAT) pathways to produce glutamate and glutamine after the successive reduction of nitrate to nitrite and nitrite to ammonia or directly through ammonium uptake (Zehr and Falkowski 1988). However, the first reduction occurs in the diatom cytoplasm via the enzyme nitrate reductase (NR) and not inside the chloroplast. It is not known if H. germanica has this enzyme but it is present in N. stella (Grzymski et al. 2002). Interestingly, nitrogen (i.e. nitrite and ammonium) assimilation by sacoglossans (e.g. Elysia viridis) was observed under light and dark conditions with significantly higher nitrogen assimilation observed under light condition (Teugels et al. 2008). The uptake of ammonium and nitrite are light dependent as their relevant enzymes require kleptoplast electron donors (NiR and GOGAT); i.e. reduced ferredoxin formed in the photosynthetic electron transport chain are used as electron donors in the reaction involving the nitrite reductase [NiR]. Furthermore, the GS metabolic reaction is ATP-dependent, and gene expression of some key enzymes (NiR, GS and GOGAT) is light regulated (Grossman and Takahashi 2001). This suggests that kleptoplasts might also have an added value in providing extra nitrogen source to metabolic pathways in foraminifera under light exposure and also possibly over short periods under dark conditions. It is also noteworthy that ammonium incorporation might take place through the glutamine dehydrogenase (GDH) pathway in the mitochondria that converts glutamate to $\alpha$-ketoglutarate, which can subsequently be assimilated in the kleptoplast via the GOGAT pathway (Teugels et al. 2008). Diatoms are also known to assimilate organic nitrogen (Antia et al. 1991), to use their ornithine-urea cycle for anaplerotic carbon fixation into nitrogenous compounds (Allen et al. 2012) and some of the benthic species present on mudflats are also able to assimilate organic carbon (Admiraal and Peletier 1979). Apparently some benthic diatoms can alternate between an auto- or heterotrophic metabolism in function of the environment. Analysing the kleptoplast DNA would provide interesting data to determine if foraminifera are capable of selecting facultative heterotrophic diatoms to improve their ability to assimilate dissolved organic compounds. Finally, another possible added value of incorporating kleptoplasts is the possibility of using them as an energy stock to be digested during food-impoverished periods particularly when foraminifera are transported below the photic zone of the sediment by macrofaunal bioturbation."

Other minor comments or corrections

Page 2 Comment: Line 12 If the authors mention "secondary role" here, then the authors need to mention about that bacteria play primary roles on carbon cycling in aerobic sediments. Reply: We agree with the reviewer and modified the sentence to clarify what we wanted to say: Line 13-16: "Their minor role in organic carbon cycling in aerobic sediments, compared to bacteria, contrasts with their strong contribution to anaerobic organic matter mineralisation (Geslin et al. 2011) and they can be responsible for up to 80% of benthic denitrification (Pina-Ochoa et al. 2010; Risgaard-Petersen et al. 2006)."

Comment: Line 16 The sentence starting with "Some benthic foraminifera: "seems appeared abruptly. Is kleptoplasty related to carbon cycling or anoxic adaptation of the foraminifera? If so, please add relevant connections from the former sentences. Reply: Agreed, since it is a separate topic and to clarify this section we changed it to a new paragraph.

Page 3 Comment: Line 13 Costal > coastal Reply: corrected

Page 4 Comment: Line 21 What is the "-" before 2.019W? Does this mean 2.019E? Reply: Corrected, the "-" was a mistake

Comment: Line 23 ∼20 kg Reply: Corrected as suggested, "±20 kg" was changed to "∼20 kg"

Comment: Line 27 Please note the filter size Reply: Added as suggested: "filtered (GF/C, 1.2 $\mu$m, Whatman) autoclaved sea-water"

Page 5 Comment: Line 11 Please explain shortly about the methods described in Jesus et al. (2008). Reply: The short explanation was already in the manuscript (type of machine, sensor position. . .), however to clarify the paragraph we added the word "concisely" to link the two sentences. "Concisely, a USB2000 (Ocean Optics, Dunedin, FL, USA) spectroradiometer with a VIS-NIR optical configuration controlled by OObase32 software (Ocean Optics B.V., Duiven, the Netherlands) was used. The spectroradiometer sensor was positioned so that the surface was always viewed from the nadir position. Foraminiferal reflectance spectra were calculated by dividing the upwelling spectral radiance from the foraminifera (Lu) by the reflectance of a clean polystyrene plate (Ld) for both of which the machine dark noise (Dn) was subtracted (eq. 1)." Comment: Line 12 50 specimens of H. germanica and : Reply: Agreed and modified : "Pigment spectral reflectance was measured non-invasively to determine the relative pigment composition on 50 fresh specimens of H. germanica, on 50 fresh specimens of A. tepida and on benthic diatom as explained in Jesus et al. (2008)."

Comment: Line 25 This is the first place appearing RLC, so please explain. Reply: Agreed, we slightly modified a sentence in the introduction to introduce the term RLC, Page 4 Line 7-9: "This non-invasive technique has the advantage of estimating relative electron transport rates (rETR) using rapid light curves (RLC) and photosystem II (PSII) maximum quantum efficiencies (Fv/Fm) very quickly and without incubation periods."

Comment: 3 X 10 specimens (or individuals) Reply: Agreed, we clarified this sentence: Page 7 line 20 "For each condition, ten specimens were used per replicate and three replicates per light treatment; furthermore all plastic Petri dishes were filled with Bourgneuf bay filtered-autoclaved seawater."

Page 6 Comment: Line 11 How long did the authors wait till the oxygen microprofiling after putting foraminifera into the tube? Reply: We added a sentence in this section to clarify this point: "Measurements were registered when the oxygen micro-profiles were stable; they were then repeated five time in the centre of the micro-tube"

Comment: Line 19 Which position of oxygen gradients were used to calculate diffusion flux? Near foraminifera? Maximum slope? Or did authors proximate in some way?

Please specify and describe. Reply: We used the $R^2$ to determine the best slope and to avoid the small O2 turbulences that often occur close to the foraminifera, therefore we modified a sentence in this section to clarify this point "The O2 concentration gradients were calculated with the oxygen profiles and using the $R^2$ of the regression line to determine the best gradient."

Page 7 Comment: Line 20 I guess Ammonia tepida exhibited chlorophyll at the start of the experiment because they still have some diatoms in food vacuoles. It may be interesting to compare the concentration of chloroplast at the begging (perhaps reflecting selective ingestion?) or reduction of chlorophyll in A. tepida as an index of degradation of chloroplast and that of H. germanica, which retain chloroplast. Reply: Thank you for the suggestion, we agree that it would be an interesting measurement but we chose to use the Fv/Fm to follow the chloroplast degradation. Although it is an indirect measurement it has the advantage of being less invasive than using microscopy, where exposure to light during the time necessary to produce an image would automatically have an impact on the chloroplast due to the microscope light.

Comment: Line 25 Please note wave length Reply: added (, excitation wave length 485 nm)

Page 8 Comment: Line 5 Triplicate measurement for each specimen? or just using 3 specimens as triplicate? Please specify. Reply: We clarified it in Page 7 line 20 "For each condition, ten specimens were used per replicate and three replicates per light treatment; furthermore all plastic Petri dishes were filled with Bourgneuf bay filtered-autoclaved seawater."

Comment: Line 9 Again, individuals or specimens are better than using "foraminifera" Reply: Agreed and replaced the word "foraminifera" by "specimens" as suggested.

Comment: Line 12 To compare the foraminifera test mean maximal elongation "between what" Reply: We clarified it in P5 line 4: "the length of the axes going from the last chamber to the other side of the test and passing by the umbilicus".

Comment: Line 23 "390 +- 42 $\mu$m (SD, n = 34)" is better Reply: Agreed and modified in the text accordingly.

Comment: Line 29 Absorption at 435 and 585 nm are not "deep absorption feature". Page 9 Comment: Line 1 In the figure 1, there is no indication of Chla at 435 nm wavelength. Reply: For the two last comments, we agree and modified the text in P9 line 20-26: "Fresh Haynesina germanica showed a typical diatom spectral signature with high reflectance in the infrared region (>740 nm) and clear absorption features around 585, 630 and 675 nm; the absorption feature around 675 nm correspond to the presence of chlorophyll a; the 585 nm feature is the result of fucoxanthin and the 630 nm absorption feature is the result of chlorophyll c (arrows, Figure 1). Ammonia tepida showed no obvious pigment absorption features apart from 430 nm (Figure 1)."

Comment: Line 3 Please note which sample (starved and kept dark under 7 days?) was used forthis spectral signatures in Figure 1. Reply: This has been clarified in P5 line 12-14: "Pigment spectral reflectance was measured non-invasively to determine and compare the relative pigment composition on 50 fresh specimens of H. germanica, on 50 fresh specimens of A. tepida and on a benthic diatom as explained in Jesus et al. (2008)."

Comment: Line 12 There is no statistical indication in Figure 3. Also, Kruskal Wallis can detect differences between several samples, but cannot say anything about the difference between s pecific two samples. Therefore, if the authors describe "Samples kept in the dark did not show an obvious decrease", then the authors need to perform another statistical analysis on this. Reply: This part is addressed in the description of fig 2 and not fig 3, there is no statistical test associated to it.

Comment: Regarding figure 3, did the authors perform any kind of "calibration" between pixel values and chlorophyll concentration? If not, the vertical axis (pixel values) does not have any numerical meaning. I therefore suggest to present as relative chlorophyll fluorescence as T0=100%. Reply: This has been clarified in the material and method P6 line 3-8: "In a RGB image each channel contains pixels between 0 and 255 values. The majority of the information regarding chlorophyll fluorescence is encoded in the red channel, therefore the green and blue channel were discarded and only the red channel was kept. The images from the different treatments were directly comparable as all images were taken using the same acquisition settings. Thus, the mean red pixel values were used as a proxy for chlorophyll fluorescence."

Comment: Line 21 No evidence of photoinhibition "of this measured range" or something Reply: Agreed and modified accordingly: "showing no evidence of photoinhibition within the light range used (Figure 4)

Comment: Line 30 "light respiration being lower than dark respiration" Based on the Table 1, LL respiration was higher than dark respiration. Does "light respiration" mean the average of LL and HL? Please specify. Reply: This has been clarified in the material and methods P8 lines 21-24: "Oxygen measurements were repeated at 300 $\mu$mol photons m-2 s-1 and in the dark at the end of the experiment (7 days of incubation) for all different light treatments (D, LL, HL) to assess the production or consumption of oxygen at these two light levels in all treatments."

Comment: Page 10 Line 2 LSD test ")". Reply: The parenthesis has been added.

Comment: Line 26 Clearly show "that"? Reply: Changed.

Comment: Page 11 Line 8 "24" umol photons? Reply: Corrected to "24".

Comment: Line 11 Do the authors have any idea on the in situ light intensity? Reply: Added "very high irradiance levels (>1000 $\mu$mol photons m-2 s-1) at the surface of the sediment during low tide". Please also note that irradiance levels are very quickly attenuated in muddy sediments. For example, at an ambient light of 1500 $\mu$mol photons m-2 s-1, light levels at 500 $\mu$m deep will be reduced to 75 $\mu$mol photons m-2 s-1 in muddy sediments with a light attenuation coefficient of 8 mm-1.

Comment: Page 12 Line 7 It seems the authors want to say "modestly" or something instead "little". Reply: "little" has been replaced by "not much"  

Anonymous Referee #2

Comment: Author analysed the functionally of chloroplast retained by some species of benthic foraminifera. Study conducted is very interesting, and the techniques used are new and applicable to other organisms, which makes the manuscript relevant to a broad readership. However, methods section needs to be carefully revised as it does not follow a logical sequence, and experimental design needs to be explained in more detail. Moreover, manuscript needs to be proofread and revised by a native English speaker. Many problems with punctuation throughout the text.

Reply: The Methods section has been carefully revised, clarified and completed.

Introduction Page 2 Comment: Ln. 19-24: Kleptoplasty is also very common in carbonate reef environments when conditions are favourable (i.e, oligotrophy; e.g., Ziegler and Uthicke 2011). Reply: We disagree with the species identification in this publication, i.e. the only kleptoplastic foraminifera mentioned, an Elphidium sp., is clearly not an Elphidium, and, therefore we prefer not to cite the publication.

Comment: Ln. 25-28: Studies by Correia and Lee need to be acknowledged and cited here as they represent a good contribution to this research field. Reply: Agreed, the studies of Correia and Lee were added at the end of this sentence.

Methods

Page 4 Comment: Ln. 11-16: Please provide a rationale for only exposing the specimens to different light levels for one week only. Reply: Agreed and clarified P7 L18-20: "A short term experiment was thus carried out (7 days) to study the effect of light on healthy specimens rather than the effect of starvation."

Page 5 Comment: Ln. 27-28: Clarify why A. tepida specimens were not starved under

light conditions, and if A. tepida was exposed to different light conditions at all. Reply: Because after one day Fv/Fm was already very close to zero. Thus all posterior measurements would be zero and meaningless.

Comment: Experimental design: Please, clarify the total number of individual used per replicate and number of replicates per treatment. Also, please clarify the experimental design. Was A. tepida exposed to different light treatments? There is no information in the methods (where it should be). It is surprising that the authors only used one paragraph to explain their experimental design, which is the most important part of the study. There is no way for the reader to know number of replicates, total number of specimens, why conditions were chosen, how light levels were reached, temperature, static or flow-through system? Detail explanation of the experimental design is necessary. Comment: Methods section does not follow a logic sequence when explaining each parameter analysed. This section needs to be carefully revised. Were all specimens used in the experiment tested for all parameters analysed? Please, clarify. Page 6 Comment: Ln. 10-11: Was one individual used at a time or all at once? Please, clarify. Comment: Ln. 20-22: What about inter specific differences? Did the authors use a pool of 7- 10 individuals for O2 consumption measurements? Or the measurements were done individually? Comment: Ln. 24: Authors stated that seven specimens were used, but previously (ln. 10) mentioned "7 to 10 foraminifera". Please, be consistent. Comment: Ln. 26: Please clarify why only two steps were used for A. tepida. Page 7 Comment: Fluoresce measurements: What light was used to measure Fo? Please, clarify Page 8 Comment: Ln. 15-16: It seems that the authors have a blocked design, but it hard to tell based on the current description of the experimental design. For example, if both species were put in the same experimental petri dish or not. That requires a more detailed description of the methods. Therefore, it is impossible to judge if authors conducted the appropriate statistical analyses. Throughout the methods section author put in brackets "3x10 foraminifera". Please, clarify if this means replicates or trials per parameter analysed.

Reply: to answer to the previous 9 comments, the Material and Methods section has been carefully revised and modified to address the different points mentioned by reviewer 2 from section 2.3 to 2.7:

[revised manuscript text omitted]

Results

Comment :Ln. 24-25 Please add ", respectively", after "This resulted in cytoplasmic biovolumes equal to 1.20 _ 107 _m3 (SD = 25 3.9 _ 106 _m3) and 1.01 _ 107 _m3 (SD = 3.65 _ 106 _m3)" Reply: Changed as suggested.

Page 9 Comment: Ln. 5-6: Figure 2 only shows data on H. germanica fluorescence. Please, amend the sentence accordingly. Reply: Corrected as requested, "Foraminiferal" was replaced by "H. germanica"

Comment: Ln. 15-19: The manuscript would improve if all these numbers were put in a table or graph. Reply: We believe it would take excessive place to include a table for only 2 values but would be happy to do it if the editor wishes.

Page 10 Comment: Ln. 20-22: Please, clarify why data is not shown. Maybe authors could add these results to supplementary material, if possible. The manuscript would benefit from a figure plotting the relative difference of Fv/Fm between light treatments, specially low and high light levels. Reply: Figure 7 already plots Fv/Fm differences between treatments; since all H. germanica treatments start from the same Fv/Fm values it is easy to compare differences between light treatments. Information about the light effect on A. tepida comes from a preliminary experiment we carried out where we saw that even low light levels would have a very strong effect on A. tepida Fv/Fm values. We decided to run the real experiment with just A. tepida in the dark because it would gives a better idea of how long the chloroplasts would be stable without any light effect, i.e. in their optimal conditions. We could repeat the experiment with just A. tepida but it seems a bit out of scope since our main objective was to investigate H. germanica and a similar experiment (i.e. with light levels) with just A. tepida would be a collection of zeros after 1-2 days.

Page 11 Comment: Ln. 7-12: Figure 4 does not show this result. Reply: The words "capture photons" were deleted from the sentence to fit better the data presented in Fig. 4: "Furthermore, H. germanica has the ability to produce oxygen from low to relatively high irradiance, as shown by the low compensation point (Ic) of 24 $\mu$mol photons m-2 s-1 and the high onset of light saturation (>300 $\mu$mol photons m-2 s-1) (Figure 4)"

Page 12 Comment: Ln. 21-23: This is expected, given that exposure to high light levels generates a lot of reactive oxygen species inside the chloroplasts. This should be mentioned and discussed. Reply: Line 21-23 concern A. tepida which was maintained in the dark, so we do not understand this comment. However, we agree that ROS could have an impact on both kleptoplasts and foramninifera. Therefore we added a sentence in the discussion P14 L26: " In H. germanica exposed to HL it is also possible that reactive oxygen species (ROS) production rates of the sequestered chloroplasts might exceed the foraminifera capacity to eliminate those ROS, thus inducing permanent damage to the foraminifera. This ROS production could also eventually damage the kleptoplasts resulting in higher kleptoplast degradation rates."

Comment: Ln. 21-23: A. tepida has no capacity to retain chloroplast according to the results, as fluoresce only persists for a couple of days, and even though some fluorescence is detected, the functionality was not analysed. Therefore, chloroplasts might be present for a couple of days, but not functional. The O2 consumption is not a proxy of functionally of kleptoplasts, and just because respiration rates were lower at 300 uE does not mean that chloroplasts were functioning. Be careful not to mix up correlation with causation. Reply: The functionality was measured using Fv/Fm. Although high Fv/Fm values are not an absolute guarantee that all photosynthetic processes are functional (e.g. the Calvin cycle) we can be sure that low or zero Fv/Fm are a result of impaired or absence of photosynthesis.

Ln. 28-32: This is very interesting. I wonder what caused this significant reduction in tolerance in these chloroplasts. Maybe the lack of a cellular protection? Would be great to see a sentence or two with thoughts from the authors of why such dramatic decrease. It would be possible that in situ the chloroplast are not functional at all. Reply: Agreed it is very interesting however we can only make suppositions and some of them are discussed in this article mainly from P14 line 1 to 10 Please also note that irradiance levels are very quickly attenuated in muddy sediments. For example, at an ambient light of 1500 $\mu$mol photons m-2 s-1, light levels at 500 $\mu$m deep will be reduced to 75 $\mu$mol photons m-2 s-1 in muddy sediments with a light attenuation coefficient of 8 mm-1.

Please also note the supplement to this comment:
http://www.biogeosciences-discuss.net/bg-2015-656/bg-2015-656-AC1-supplement.pdf

**Supplement:**

[revised manuscript text omitted]

---

## Author Response (AR2)

Dear Dr. Middelburg,

Please find enclosed our revised manuscript entitled "**Effect of light on photosynthetic efficiency of intertidal benthic foraminifera**" by Thierry Jauffrais, Bruno Jesus, Edouard Metzger, Jean-Luc Mouget, Frans Jorissen, Emmanuelle Geslin, for submission to Biogeosciences as an original research paper.

We agree with the different comments you have done and modified the manuscript as requested.

We hope that these revisions will fulfill all the requests for publication.

Thank you for your work,
Best regards,
* * *
Thierry Jauffrais
* * *
Dr. Thierry Jauffrais
UMR CNRS 6112 LPG-BIAF, Université d'Angers, UFR Sciences, 2 Bd Lavoisier, 49045 ANGERS CEDEX 01, France
Thierry.jauffrais@univ-angers.fr
TEL: +332 41 73 50 09

Comment:
- do not use forward referencing: i.e. delete Cesbron et al. (submitted) in the text, p.2, p.3.
Reply:
- Agreed and modified the forward referencing as Cesbron pers. Comm.; the reference of the submitted manuscript was also removed from the reference section.

Comment:
- p. 8, line 25: For all conditions..... Fv/Fm was measured.... period and measurements were done...
Reply:
- Agreed and modified as suggested

Comment:
- p12: line 28: local biogeochemical processes...
Reply:
- Agreed and added as suggested

Comment:
- p14, line 27: replace HL with high light
Reply:
- Agreed and modified as suggested

Comment:
- p16, line 6: why is nitrogen suddenly entering the stroy here?
Reply:
- Agreed the sentence about nitrogen was removed from the conclusion. It was here to highlight a section on nitrogen assimilation that has been suppressed from the manuscript during the reviewing process.

[revised manuscript text omitted]